# Charge-Discharge Characteristics of Textile Energy Storage Devices Having Different PEDOT:PSS Ratios and Conductive Yarns Configuration

**DOI:** 10.3390/polym11020345

**Published:** 2019-02-16

**Authors:** Ida Nuramdhani, Manoj Jose, Pieter Samyn, Peter Adriaensens, Benny Malengier, Wim Deferme, Gilbert De Mey, Lieva Van Langenhove

**Affiliations:** 1Department of Materials, Textiles, and Chemical Engineering, Centre for Textile Science and Engineering, Ghent University, B-9000 Gent, Belgium; Benny.Malengier@UGent.be (B.M.); Lieva.VanLangenhove@UGent.be (L.V.L.); 2Department of Textile Chemistry; Polytechnic STTT Bandung, Bandung, Jawa Barat 40272, Indonesia; 3Institute for Materials Research (IMO), Hasselt University, B-3590 Diepenbeek, Belgium; manoj.jose@uhasselt.be (M.J.); pieter.samyn@uhasselt.be (P.S.); peter.adriaensens@uhasselt.be (P.A.); wim.deferme@uhasselt.be (W.D.); 4IMEC vzw–Division IMOMEC, Wetenschapspark 1, B-3590 Diepenbeek, Belgium; 5Department of Electronics and Information Systems, Faculty of Engineering and Architecture, Ghent University, B-9000 Gent, Belgium; Gilbert.Demey@UGent.be

**Keywords:** PEDOT:PSS, conductive polymer, textile energy storage device, solid electrolyte, stainless steel, Ag/PBO

## Abstract

Conductive polymer PEDOT:PSS, sandwiched between two conductive yarns, has been proven to have capacitive behavior in our textile energy storage devices. Full understanding of its underlying mechanism is still intriguing. The effect of the PEDOT to PSS ratio and the configuration of the electrode yarns are the focus of this study. Three commercial PEDOT:PSS yarns, Clevios P-VP-AI-4083, Ossila AI 4083, and Orgacon ICP 1050, as well as stainless steel and silver-coated polybenzoxazole (Ag/PBO) yarns, in various combinations, were used as solid electrolytes and electrodes, respectively. Analyses with NMR, ICP-OES, TGA, and resistivity measurement were employed to characterize the PEDOT:PSS. The device charge-discharge performance was measured by the Arduino microcontroller. Clevios and Ossila were found to have identical characteristics with a similar ratio, that is, 1:5.26, hence a higher resistivity of 1000 Ω.cm, while Orgacon had a lower PEDOT to PSS ratio, that is, 1:4.65, with a lower resistivity of 0.25–1 Ω.cm. The thermal stability of PEDOT:PSS up to 250 °C was proven. Devices with PEDOT:PSS having lower conductivity, such as Clevios P-VP-AI-4083 or Ossila AI 4083, showed capacitive behavior. For a better charge-discharge profile, it is also suggested that the PEDOT to electrode resistance should be low. These results led to a conclusion that a larger ratio of PEDOT to PSS, having higher resistivity, is more desirable, but further research is needed.

## 1. Introduction

For the last decade, conductive polymer from poly(3,4-ethylenedioxythiophene):poly(4-styrenesulfonic acid), which is also popularly known as PEDOT:PSS, has attracted a great interest and been the subject of many intensive studies owing to its many potential future applications for fabrication of low cost, flexible, and printable electronic devices [1]. It is considered to be one of the most technologically important electronic materials due to its simultaneous excellent properties, such as high electrical conductivity, processibility, high stability, flexibility, and good transparency. These exceptional properties led this polymeric system to be used in a large variety of applications, such as polymeric anodes for organic photovoltaics [2,3], light-emitting diodes [4,5,6,7], flexible electrodes [8,9], supercapacitors [10,11,12,13], electrochromic devices [14,15], field-effect transistors [16,17], and antistatic coatings [18]. 

From its general chemical structure, it can be observed that conducting PEDOT can be electrostatically bound to the PSS polyanion. PEDOT itself is an insoluble material, but when it is synthesized in the presence of PSS, an aqueous dispersion is obtained and can be cast into thin films [19]. The polyelectrolyte complex is available commercially in the form of stable polymer dispersion in water with a variety of solids content and ratio of PEDOT and PSS, which translates to different levels of conductivity [20,21]. Different applications require different levels of conductivity. For example, antistatic coating applications require high conductivity for the PEDOT:PSS layer whereas in organic light-emitting diode (OLED) applications, lower conductivity for PEDOT:PSS is required [22]. In this regard, it is interesting to note that PEDOT:PSS films demonstrated different conductivities depending on the direction of transport of charges, hence anisotropic conduction. Nardes et. al. [23,24] found that in- and out-of-plane conductivity of PEDOT:PSS with a large ratio of PEDOT and PSS showed a large anisotropy, roughly by a factor of 400, which was explained by the corresponding anisotropy in the granular structure of the film. This observation is particularly important when one considers a different configuration for the device. In typical state-of-the-art organic solar cells and light emitting diodes, the main role of PEDOT:PSS is to transport charge in the out-of-plane direction because it acts as an intermediate layer in a sandwich structure between two working electrodes. This is not the case with our device, where the two electrodes are confined laterally along the plane of the film, with charges being transported most likely in the in-plane direction.

Studies on PEDOT:PSS have largely focused on its electrical conductivity [25,26,27,28,29,30], which was particularly driven by its use as an electrode in electronic devices. As mentioned in the literature [21,24,29,30,31,32], the electrical conductivity of the films strongly depends on the fabrication process and morphology, which suggests that the strategy to increase conductivity rests on the optimization of the film morphology and micro/macro structure. For this reason, the understanding of the structure of PEDOT:PSS in the solid film and its dispersion in water is crucial for improving the film fabrication process and tapping the strategies to improve the overall performance of electronic devices that employ this particular polymeric system. Quite different from the majority of applications, PEDOT:PSS has been used in textile-based energy devices as a solid electrolyte layer that covered two parallel electrode yarns made of stainless steel and/or silver coated polybenzoxazole (Ag/PBO) [33,34]. The device has been successfully tested in the laboratory and consistently stored charges with a voltage of up to 1.2 V after being charged at 3.0 V for at least 40 min. The device has also passed a series of charge-discharge tests without showing any sign of deterioration over time, demonstrating it as rechargeable [33,35]. The central issue and challenge at this stage of the development are how to improve the charging and storage capability of the device. In the charging of the device, for example, we have not been able to reach the full voltage delivered by the power supply and can only attain approximately 40–60% of the input voltage. Secondly, when disconnected from the power supply, the voltage dropped almost instantly to more than 50%, showing a sign of serious self-discharge or current leakage, and only then a slow decay followed that took weeks to drain all the charges. In the most practical terms, our ambition is to have an energy storage device that can be charged fully to the level of the input voltage and has a longer time of operation. In other words, we aim to improve the capacitance of the device. This requires an understanding of how charges are stored in the device and how each of the components, as well as design configuration, come into play in the mechanisms. 

With regard to the different ratios of PEDOT and PSS in the complex polymer of PEDOT:PSS, in this study, we were particularly interested in how the resulting different conductivities affect the working and performance of our textile-based energy storage device. Based on the previous studies [34,36], we believed that the storage mechanism in the device was based on pseudocapacitor, which involves both charge separation and mixed ionic and electronic transport. For that reason, it was quite plausible that the performance of the device is affected by the conductivity of PEDOT:PSS film to give different storage capability for each of the devices. With this study, we expect to obtain more information for further elucidation of the storage mechanisms of the device. In order to support the latter, the investigation was conducted with different design configuration involving the use of symmetrical and asymmetrical electrode yarns from inert and non-inert materials.

## 2. Materials and Methods

### 2.1. Materials

The textile energy storage devices developed for these experiments generally consisted of three parts: a textile substrate, conductive yarns which acted as electrodes, and the electroconductive polymer which functioned as a solid electrolyte. A twill woven polyester–cotton fabric with a warp and weft density of 42 yarns/cm and 29 yarns/cm, respectively, obtained from our laboratory was used as a textile substrate. NK-Guard NDN-7 ENQ (Indonesia Nikka Chemicals Co. Ltd., Karawang, Indonesia) was utilized as a fluorocarbon-based water repellent (WR) agent. A layer of thermoplastic polyurethane (TPU) obtained from Epurex Film Company (Bomlitz, Germany) was employed to mask the surface and create the cell area. Two types of conductive yarn, pure stainless steel Bekaert Bekinox (Bekintex NV, Wetteren, Belgium) and silver-coated polybenzoxazole (Ag-PBO) of Amberstrand (Syscom Advanced Materials Inc., Columbus, OH, USA), were used as electrodes with different pair combinations as described further in the next section. As the solid electrolyte, three commercial brands of PEDOT:PSS dispersion were employed and applied individually in each device. They are (1) M121 PEDOT:PSS AI 4083 (Ossila Ltd., Sheffield, UK); (2) Clevios P-VP-AI-4083 (Heraeus, Hanau, Germany), and (3) Orgacon ICP 1050 (Sigma Aldrich, Darmstadt, Germany). The resistivity of each PEDOT:PSS layer on the glass at 2.5 × 2.5 cm^2^ was measured by a two points measurement method. 

### 2.2. Methods

#### 2.2.1. Fabric Pre-Treatment with Water Repellent Agent

Before being used as a textile substrate for the device, the fabric was pre-treated with NK-Guard NDN–7 ENQ water repellent agent (fluorocarbon-based) to impart a hydrophobic effect on the fabric surface, and therefore prevent absorption of PEDOT:PSS into the fabric structure giving a confined PEDOT:PSS distribution within the cell area of the device. Two percent of the solution was first impregnated onto the fabric by the padder set at 80% WPU (wet pickup) and then dried with stenter at 120 °C for 2 min. The drop test with water and PEDOT:PSS performance on the fabric showed an effective hydrophobicity as shown in Figure 1.

#### 2.2.2. Fabrication of Devices

Nine devices with different combinations of types of electrodes and electrolyte plus three so-called blank devices (see Table 1) were prepared on 5 × 5 cm^2^-sized water repellent (WR)-treated fabric. The cell area, which comprised of a pair of electroconductive yarns and drop-coated PEDOT:PSS film, was created by masking a layer of TPU film (thickness of 25 µm) to the fabric by a heat press at 100 °C for 2 min. The TPU film had at its center a rectangular hole of 6 mm to 10 mm to create an unmasked region and a confined area for the subsequent drop coating of PEDOT:PSS polymer dispersion. Before the drop-coating procedure, a pair of electrode yarns, as set for each device (Table 1), was stitched at a distance of 1.5 mm apart in the middle of the cell area (See Figure 2b). An initial test by multimeter was performed on each device to make sure that there was no short circuit due to the contact between the stitched yarn electrodes. The drop coating of PEDOT:PSS was performed as described in the previous work [36]. However, in this study, a slight technical modification was done for the PEDOT:PSS coating method due to the hydrophobic surface properties of the fabric leading to having less amount of PEDOT:PSS polymer used for each device. With the perfect hydrophobicity on the fabric surface, there was no diffusion of the diluted polymer into the pores of the fabric as visually observed in the previous devices [33,34,35], and therefore, all polymer dispersion dropped into the cell surface could be completely distributed on the surface of the cell area. This way, instead of having seven drops, as described in the previous works, five drops were sufficient to cover the cell area. As performed previously [34], drying of film was attained in the oven at 100 °C for 10 min layer by layer between each drop. During the coating procedure, the polymer dispersion was kept in the oven in order to make it more concentrated over time. After the fourth drop, the polymer dispersion had formed a paste, and thus, for the last coating, the paste was simply swept by a mini squeegee to improve the distribution. An example of the actual device can be seen in Figure 2.

#### 2.2.3. Determination of PEDOT:PSS Ratio by NMR and Sulfur Content by ICP-OES

^1^H NMR (Nuclear Magnetic Resonance) spectra, to determine the composition of PEDOT and PSS in the PEDOT:PSS polymer dispersion, were recorded on a 400 MHz (9.4 Tesla) Inova spectrometer (Agilent Technologies Inc., Santa Clara, CA, USA) using a 5 mm four-nucleus PFG (pulsed-field-gradient) probe. The PEDOT:PSS dispersion was diluted with deuterium oxide (D_2_O). The chemical shift scale (δ) in ppm was calibrated relative to TSP-d4 (3-(trimethylsilyl)-2,2,3,3-tetradeuteropropionic acid sodium salt) as a chemical shift reference at 0 ppm. Free induction decays were collected with a 90° pulse of 6.0 µs, a spectral width of 6 kHz, an acquisition time of 4 s, and a preparation delay of 12 s. Each free induction decay was zero-filled to 65 K points and multiplied by a 0.5 Hz exponential line-broadening function prior to Fourier transformation. The elemental analysis to determine the concentration of the sulfur (S) content in each of the PEDOT:PSS samples was performed by ICP-OES (inductively coupled plasma-optical emission spectroscopy). The system consisted of an Optima 8300 (Perkin Elmer, Waltham, MA, USA) and an S10 autosampler equipped with the Syngistix software version 2.0.0.2236. Emission lines of the Sulfur were monitored at the wavelengths of λ= 181.975 nm, 180.669 nm, and 182.863 nm and averaged. Materials prepared for the measurement were 1000 mg/L S (sulfur) standard solution purchased from Merck KGaA (Darmstadt, Germany), 69.0–70% nitric acid (HNO_3_) purchased from J.T. Baker (Deventer, The Netherlands), and MilliQ water (arium 611UV, Sartorius, Gottingen, Germany). The sample of PEDOT:PSS was diluted in 100-fold with 0.5% HNO_3_ and mixed thoroughly prior to the ICP-OES analysis. Sulfur calibration standards were prepared by serially diluting the standard S working solution with 0.5% HNO_3_, resulting in a concentration of 1, 5, 10, 50, and 100 mg/L S. Each sample was prepared in 3 replicates.

#### 2.2.4. Thermogravimetric Analysis of PEDOT:PSS 

Thermogravimetric analysis was conducted to obtain particular information on water uptake of PEDOT:PSS, that is, the number of bound water molecules in the film. 50 mL of each PEDOT:PSS dispersion was freeze-dried for 24 h by an Eyela FD-5N freeze dryer (Tokyo Rikakikai Co. Ltd., Tokyo, Japan), and then stored in the desiccator to make sure that it did not take up any extra water or moisture and remained dry while in storage before the thermogravimetric analysis (TGA). TGA was performed on a Diamond TG/DTA instrument by Perkin Elmer (Shelton, CT, USA) operating from 30 to 600 °C at a heating rate of 5 °C/min under the flux of nitrogen at 20 mL/min. For the TGA measurements, 0.897, 1.751, and 2.177 mg PEDOT:PSS sample each from Orgacon, Ossila, and Heraeus, respectively, were taken and placed in an aluminum pan for the analysis.

#### 2.2.5. Electrical Characterization: Charge-Discharge Test 

Charge-discharge characteristics of all devices were measured by using a microcontroller “Arduino/Genuino-Uno”, which was set up specifically for this particular purpose with the circuitry shown in Figure 3. The charging of the device was done by switching to position A which supplied the device with a DC voltage (*V_s_*) of 3 V with an external resistance of 33 kΩ (*R*). The charging process was stopped by switching to position C after 1500 s when the voltage reading (*V_c_*) reached a maximum and remained stable at that level. This charging time was taken as our standard charging time based on our preliminary study. Discharging took place with the same external resistance of 33 kΩ (*R*). During the process of charging and discharging, the output voltage (*V_c_*) was recorded automatically by the program. The position of the device is presented by the name TESD, which stands for the textile energy storage device. The hypothetic details of the schematic circuit of the TESD cell are presented in Section 3.3 of this paper.

## 3. Results and Discussion

As aforementioned, the fabric used as a textile substrate for devices in this research was pre-treated with a fluorocarbon-based water repellent agent to give a hydrophobic surface. The idea behind the water repellent-pretreatment was to prevent the drop of PEDOT:PSS dispersion from penetrating through and migrating across the fabric structure to allow it to get collected in a controlled amount and distribution within the cell area during the drop-coating process (see Section 2.2.2). Practically, the procedure was our attempt to control the mass loading of PEDOT:PSS during the process of coating, and reproduce it for the quantitative comparison while evaluating device performance. Additionally, we were also interested in different approaches and strategies to control and tune the performance of our device. We believe that the ability to confine the formation of a film within a predetermined cell area during the drop coating process can be critical to the performance of a device and facilitate the more quantitative analysis on molecular interactions between the electrode yarns and the PEDOT:PSS. The contact area between PEDOT:PSS and the electrode yarn may have changed as well. However, this was not the focus of this study, so we do not present any data or analysis regarding the distribution and molecular or electronic interaction of PEDOT:PSS in the cell area. As shown in Figure 2, as a result of water repellent-pretreatment, PEDOT:PSS coating became concentrated within the cell area whereas in the devices reported previously [20,33,34,35,36], the polymer migrated out of the cell area. It demonstrated that the technique could be used to control the loading and distribution of PEDOT:PSS polymer dispersion during the coating process. 

The results and discussion about the effect of PEDOT:PSS ratio on the performance of the energy storage devices, which is the main focus of this study, are presented in the following sections. A preliminary charge-discharge measurement to compare the performance of devices using WR-pre-treated and non-WR-pre-treated fabric was carried out beforehand to obtain the general result. The devices exhibited more or less identical charge-discharge behavior. The devices measured were made of PEDOT:PSS Ossila AI 4083 and a pair of stainless steel electrode yarns. As shown in Figure 4, during the charging process (3 V for 1000 s), in general, the voltage of both devices increased gradually and dropped sharply just after the charging stopped. The stored charges then dropped slowly and remained stable for some time during the discharge time. However, the charge-discharge voltage levels were slightly different, where the device with pre-treated fabric showed approximately 20% higher level of discharge voltage during the equilibrium of discharging time (between 1000–2000 s) as shown in Figure 4 below. 

### 3.1. PEDOT to PSS Ratio and Sulfur Content 

Proton (^1^H) NMR was employed to analyze the ratio between PEDOT and PSS contained in each type of commercially available PEDOT:PSS polymer dispersions used in this research. The ^1^H NMR spectra presented in Figure 5 showed the peaks in the spectral area between 3–4 ppm (PEDOT) and 0–2 ppm (PSS), which are the characteristics of PEDOT:PSS. During the analysis, we found that Clevios and Ossila exhibited identical spectra, which gave an indication that they might be of similar composition. This was further confirmed by their respective technical information as shown in Table 2. On the other hand, slight but significantly different peaks were observed for Orgacon, particularly in the spectral area of 3–4 ppm, leading to different results of the ratio of PEDOT and PSS. Calculation of the area under each of the above-mentioned peaks gave quantitative information regarding weight and molar ratios of the three commercial brands of PEDOT:PSS used in this research. From Table 2, the molar and weight ratio of PEDOT and PSS in the polymer dispersion of Clevios and Ossila is 1.00:5.26 and 1.00:6.92, respectively, whereas for Orgacon, it is 1.00:4.65 and 1.00:6.11. From this analysis, except for a slight difference in sulfur content, it can be concluded that Clevios P-VP-AI-4083 and Ossila AI 4083 are identical. Orgacon ICP 1050, on the other hand, is different and has a smaller average number of PSS units electrostatically attached per PEDOT polymer chain, which is also evident from the lower sulfur content measured by ICP-OES. At this point, it can also be seen that the differences in PEDOT:PSS ratio between the three commercial samples are in agreement with their resistivity, and hence conductivity, although the ratio is not the only factor. The other factors like additional solvent and secondary dopants can also affect the conductivity. However, the fact that different performance of devices using each type of PEDOT:PSS, discussed in Section 3.3, tells us that other factors like different batch and storage time may play an important role as well. From the measurement, both Clevios and Ossila have the same resistivity ρ of 1000 Ω.cm, which is obviously much larger than that of Orgacon’s, 0.25–1 Ω.cm. Our measurements are in reasonably good agreement with the technical data provided by each of the manufacturing companies. As shown by the data, what appears to be the only slight difference in PSS content turns out to be highly important and has a tremendous effect on the electronic properties of PEDOT:PSS, which in this particular case is their resistivity.

As understood from the literature [21], the conductivity of PEDOT:PSS has a direct relation with the ratio of PEDOT to PSS because of the way the intrinsic conductivity of this polymer is formed. With the same amount of PEDOT molecules, the larger proportion of PSS in the polyelectrolyte system leads to lower conductivity. This can be explained by the fact that with more polyanionic PSS in the polymer, the density of charge transporting PEDOT sites are lower. The PSS itself is known to not directly support the charge transport [21]. With its highly acidic and polar sulfonic group, it functions more as a counter ion to form a stable dispersion in the system of polyelectrolyte complex paired with PEDOT. However, the excess amount of PSS, especially in the non-stoichiometric ratio of PEDOT:PSS, in water leads to the formation of an intramolecular interaction between the PEDOT and PSS molecules in addition to the Coulomb interaction between anionic ions from PSS with the free cationic ions in the PEDOT, creating a scramble egg-like arrangement of the molecule mixture [21,37]. As a consequence, the mobility of the conjugated charge carrier decreased accordingly leading to much higher resistivity of 1000 Ω.cm (compared with only 0.25–1 Ω.cm for Orgacon) and lower conductivity for both Clevios and Ossila. As described in the later sections of this study, this difference in resistivity and conductivity has significant consequences to the behavior and performance of the devices made of each of the Clevios and Ossila on one hand and Orgacon on the other hand. A similar phenomenon has been observed in semiconductor compounds like GaAs, InP, CdS, CdSe, etc. For GaAs, for example, the intrinsic material has a very low electric conductivity [38]. A small excess of Ga converts the GaAs into an n-type semiconductor, with increased electric conductivity. An excess of As gives rise to a p-type semiconductor with an increased conductivity as well [38]. Our discussion centers and focuses on explaining why and how the two groups of PEDOT:PSS differ in their behavior and performance.

### 3.2. Thermal Characteristics of Each PEDOT:PSS

The TGA (thermal gravimetric analysis) was employed to analyze the water uptake of PEDOT:PSS film that functions as the solid electrolyte for the device developed in this research. The analysis is of importance in relation to the discussion about the conductivity of the polymer. By this method, we can determine not only the amount of absorbed water but also the adsorbed or bound water molecule in the internal structure of PEDOT:PSS film, which has been known to be highly hygroscopic polymer due to the presence of sulfonic acid groups. It was suspected that the presence of bound water molecules might have an effect in increasing the conductivity of the polymer. Figure 6 presents the TGA curves of the Ossila, Orgacon, and Clevios PEDOT:PSS used in this research.

It can be seen from the curves that significant weight loss occurred during the first stage of heating from 30 to 100 °C, which was originated from the evaporation of absorbed water. Based on the exact amount of weight loss at 100 °C taken from the data obtained, it was calculated that the water content of the three polymers after being freeze-dried was approximately 14.14%, 14.37%, and 17.42% for Orgacon, Ossila, and Clevios, respectively. On further heating to 250 °C, there was no further weight loss indicating the absence of adsorbed and a bound water molecule in the internal structure of PEDOT:PSS films under investigation. PEDOT:PSS has been known to have very good thermal stability [21,39]. It can withstand heat treatment up to 250 °C without considerable decomposition. However, a gradual decrease of weight appears on further heating to 350 °C, which can be considered as an indication for the breaking up of the PSS sulfonate groups leading to a weight loss of approximately 30%. Elschner et al. [21] explained a similar phenomenon in their TGA results of PEDOT:PSS at 100–250 °C. They reported the fragmentation of the sulfonate group, which was indicated by the increase in the ion currents because of the presence of the resulting SO_2_ ions. As the PSS proportion in Orgacon is less than those in Clevios and Ossila, the loss of weight due to the destruction of the PSS sulfonic group is lower as shown by the curve in Figure 6. Furthermore, significant decreases occurred at temperatures between 350–450 °C. Weight loss of up to approximately 70%, 74%, and 78% was exhibited by Orgacon, Ossila, and Clevios, respectively. This indicated the oxidation of sulfur or carbon atoms in the thiophene rings which possibly led to the formation of a nonconducting sulfoxide and sulfone structures or hydroxyl group followed by rearrangement into its oxide form [21]. From the TGA curves, it can be concluded that the three PEDOT:PSS’s have good thermal stability up to 200 °C. In this respect and in relation to the heat treatment of PEDOT:PSS in the fabrication of our devices, the only heating applied was the repeated drying at temperatures between 90–110 °C during the drop-coating of PEDOT:PSS dispersion into the cell area. Looking at the TGA data, the heat treatment should not disturb the integrity of the PEDOT:PSS polymeric structure nor its conductivity, although the heating was carried out repeatedly. The DTA curves presented in Figure 6 give further information on the total degradation of the polymer that occurred at T = 450 °C, where the exothermic event took place with the highest amount of energy. 

The presence of sulfonic acid groups in PEDOT:PSS represents the strong hygroscopic characteristic, which allows the formed film to easily take up moisture in the ambient condition [21]. Based on visual observation, the PEDOT:PSS film on our device did not seem to undergo visible swelling under ambient condition during the three months storage. However, there was a consistent change in the charge-discharge performance: the voltage level of charging increased while the discharging voltage level decreased with storage time (Figure 7). Graphs in Figure 7 show the results from a series of consecutive measurements of a device made of PEDOT:PSS Ossila AI 4083 and stainless steel electrode yarns. The device was first measured just after it was made, week 0, and then it was stored in room conditions for the next measurements after every two weeks until week 12. The charge-discharge voltage was measured with the Arduino-Uno (charging at 3 V for 1500 s and discharging for another 1500 s).

The increase and decrease of charging and discharging voltage described above, while the load of the external resistance remained the same (i.e., 33 kΩ throughout the experiments), can be taken as an indication of increasing internal resistance of the device over time. The larger internal resistance had caused the charging voltage to increase. However, when discharging, assuming that the TESD had the same voltage after being charged, the device needed to overcome the internal resistance which then led to the observation of lower discharging voltage. It is still questionable, though, as to what has caused this increase in the internal resistance of the device. The fact that PEDOT:PSS is highly hygroscopic could be one of the reasons. Water uptake during the storage could have caused some changes in the internal structure leading to a decrease in the mobility of charge carriers which then appeared as an increase in internal resistance. Further investigations are certainly needed to corroborate the above-suggested explanations.

### 3.3. Charge-Discharge Characteristics

Charge-discharge characteristics of each device using combination of electrodes with different types of PEDOT:PSS, as listed in Table 1, were measured and then compared, especially to see the effect of the ratio of the PEDOT to PSS as well as the influence of the electrode material, in the equal devices, on the charge-discharge performance of each device. The graphs in Figure 8 present the charge-discharge characteristics of devices A1, B1, C1 (Figure 8a) and A2, B2, C2 (Figure 8b), where the symmetrical SS/SS and Ag-PBO/Ag-PBO electrode yarns were used, respectively. On the other hand, asymmetrical SS/Ag-PBO electrode yarns were employed to the devices A3, B3, and C3 and their charge-discharge measurement results can be seen in Figure 9. The different electrical potential was applied during the measurements of these devices. The positive potential was applied to the SS electrodes first, and therefore the Ag-PBO electrodes had a negative potential (Figure 9a). Given one week to empty the charge in the devices, the opposite measurements, where the positive potential was given to the Ag-PBO electrodes, were then carried out (Figure 9b). Each device was charged at 3 V for 1500 s, then subsequently followed by discharging right after the charge was stopped, and the decay of charges initially stored in the device was observed for another 1500 s. As can be seen, the charge-discharge profiles presented in Figure 8 and Figure 9 are mostly noisy. The noise was found in all measurements using Arduino/Uno. As the curves still show the typical charge-discharge profile of our device, the noise is neglected in our discussion. In addition, it is important to note that according to the previous publication [36], the noisy curves were observed from the devices with a thin layer of PEDOT:PSS film in the cell surface. The mobile ions inside the PEDOT:PSS gave an inevitable rise to noise. The ions being a polymer, and hence quite large, create the typical shot noise. By evaluating the autocorrelation function of the noise, it was proven that we were dealing with the shot noise. The PEDOT:PSS layer was thick and covered the cell surface to avoid the noise, but it is still shown. So, we believe that the noise mostly comes from the shift of measurements.

It can be seen from the graphs in Figure 8 and Figure 9 that different PEDOT:PSS polymers showed different charge-discharge profiles, especially for those using Orgacon compared to Ossila and Clevios. In all charge-discharge measurements, regardless of the electrode yarns used and their set of combinations, devices using Orgacon as a solid electrolyte showed zero levels of charge and discharge voltage at all observation times. The discharge profiles from these results are similar to those found in the previous works of Sheila et al. [33]. As discussed earlier, based on our characterization and analysis using NMR and ICP-OES, the Orgacon ICP 1050 was found to be of low resistance and thus, the high conductive type of PEDOT:PSS, which for the most part stems from the lower proportion of PSS in the mixture as compared to Clevios and Ossila. 

The zero charge-discharge profile of devices having Orgacon PEDOT:PSS, as observed in Figure 8 and Figure 9, suggested the absence of capacitive behavior in the system, which resulted in the absence of charging and discharging voltage. The polymer of Orgacon ICP 1050, which is a very good conductor, acted as a resistor with very low resistivity, even with a short circuit, while the electrodes themselves were still perfect. With this assumption, a schematic circuit depicting the measurement of Orgacon device showing no capacitive behavior is presented in Figure 10. It should be noted that the circuit is only a simplification of the system, where the device behaved as a wire instead of a TESD. In the simplified circuit, *R* is the external resistance of the system, and *V_s_* and *V_r_* are the source and device output voltages, respectively. Here, the device is represented only by its internal resistance *r.* By following the Kirchhoff’s Law, as the total potential difference in the system is zero, ∑V=0, the mathematical relationship between all the elements can be derived using the following formula: (1)$$I\left(R+r\right)-V_{s}=0$$
(2)$$I=\frac{V_{s}}{R+r}$$
(3)$$V_{r}=I.r$$
(4)$$V_{r}=\frac{r}{R+r}V_{s}$$

Note: the TESD is only represented by the presence of *r and V_r_* which is practically zero in value.

From (4), when *r* << *R,* the device voltage *V_r_* will be close to zero regardless of how much voltage is supplied to the system by the power source *V_s_*. This explains our observation on flat and, practically, zero charging of the devices made up from Orgacon ICP 1050. The complete opposite charge-discharge profiles should be obtained on blank devices, as then no current can flow between the electrodes (infinite resistance). This is presented in the Appendix A, which shows that the absence of PEDOT:PSS solid electrolyte in the devices caused no current flow with an immediate voltage over the electrodes as applied by the source. 

On the other hand, devices with PEDOT:PSS having Clevios VP-AI-4083 and Ossila AI 4083 both showed capacitive behavior, represented by the charge-discharge profiles as shown in Figure 8 and Figure 9. In general, all devices showed almost identical charge-discharge profiles, except the different combinations of electrode yarns with the same type of PEDOT:PSS, Clevios or Ossila, resulted in different charge-discharge levels. This indicates that we need to consider the electrode-PEDOT resistances in the forward (current moving to PEDOT) and backward (current moving out of PEDOT) configuration. For the same polymer coating, the charging capacity should be equal, which can be represented by an internal capacitance and equivalent series resistance (ESR) of the device, as well as an internal discharge. Figure 11 presents the possible configuration of the TESD cell having Clevios and Ossila PEDOT:PSS, which consistently shows capacitive behavior.

As can be seen in Figure 8a,b, devices using a pair of Ag/PBO electrode yarns exhibited higher charging voltage at 1500 s than those using SS electrode yarns, but lower discharging voltage. Following our previous argument in Section 3.2, this means that the internal resistance of our TESD is higher when using Ag/PBO yarns as the electrode material. As the only difference is the electrode, this indicates that Ag/PBO has a higher resistance than SS yarns. It can be observed that Figure 9a shows a similar discharge behavior as in Figure 8a, whereas Figure 9b is similar to Figure 8b. Comparing the two sets of figures provides an explanation of the different behavior caused by the use of different electrode materials as well as the role of their internal resistance to the performance of the device. In the devices having an asymmetrical configuration of electrodes SS/Ag-PBO, the electrode resistance of current flow from PEDOT to Ag was apparently high. As current flows from positive to negative and electrons in the opposite direction, when SS was set to positive (+) and Ag-PBO negative (−) (Figure 9a), electrons needed to move from Ag into PEDOT (+) sites. This also explains the high charging voltage in Figure 9b. When SS was set as the negative electrode, the electrons also needed to move at the Ag electrode and then to move into PEDOT (+) sites. The inverse, current flowing from Ag to PEDOT (so electrons from PEDOT to Ag to form a PEDOT+ site), is not problematic as the observed discharge and charge are then comparable to the SS electrodes.

In general, the charge-discharge profile of devices with Clevios and Ossila is identical, although, from all measurements, Clevios always showed slightly higher charging voltage than Ossila, and a bit lower discharging voltage, indicating a somewhat higher internal resistance. The weight and molar ratios of both are exactly similar, hence, their conductivities. Based on our characterization and analysis, the only different characteristic between the two polymers is that Clevios has less sulfur content than Ossila. This might have relations with the higher internal resistance as explained earlier, although more investigations are needed to conclude. It is also important to mention that our PEDOT:PSS-based device is a non-linear component, that is, the stored charge is not proportional to the applied voltage, so the capacitance value is not defined. However, it is possible to define an “equivalent” capacitance by the following Equations (5) and (6), assuming for the device charged at 3 V, as detailed in our previous report [40].
(5)$$\frac{1}{2}C_{eq}V_{0}^{2}=E_{el}=0.8101\;mJ=0.22\;µWh$$

The value of $E_{el}$ was known to be 0.8101 mJ = 0.22 µWh by the calculation described in the paper [40], and therefore, the value of the equivalent capacitance can be calculated as follow:(6)$$C_{eq}=\frac{2E_{el}}{V_{0}^{2}}=180\;µF$$

## 4. Conclusions

Energy storage devices using three different commercial brands of PEDOT:PSS, Clevios P-VP-AI-4083, Ossila AI 4083, and Orgacon ICP 1050, as the solid electrolyte, were assembled especially to study the effect of the ratio of PEDOT to PSS on their charge-discharge performances. As electrodes, we used stainless steel (SS) and silver-coated polybenzoxazole (Ag-PBO) yarns, with various combinations: symmetrical SS/SS and Ag-PBO/Ag-PBO as well as asymmetrical SS/Ag-PBO. From the NMR analysis, we found that the Clevios and Ossila are almost identical, as evidenced by the similar NMR spectra as well as by their molar and weight ratio, that is, 1.00:5.26 and 1.00:6.92, respectively. Utilization of these two types of PEDOT:PSS gave good charge-discharge profiles of the devices. Otherwise, devices with the Orgacon showed no charge-discharge profiles. Its PEDOT:PSS molar and weight ratios are 1.00:4.65 and 1.00:6.11, respectively. The lower proportion of PSS leads to the higher conductivity of the polymer. From the results, it can be concluded that for our specific developed energy storage devices, PEDOT:PSS with lower conductivity is more suitable, and the ratio significantly affects the conductivity. It should be investigated if a further increase in the ratio would lead to further benefits. It can be also concluded that the resistance of electrode-PEDOT must be low, which is not the case for Ag when used as the positive electrode. In addition, the lower sulfur content seems to improve the results as shown by Ossila A1 4080 that has the best discharge properties. This knowledge leads to a better understanding of the underlying mechanism of the devices. Further work is needed to elaborate on the effect of the ratio by modifying the doping process or with secondary doping. This finding also gives directions into the improvement of devices’ performance with higher capacitive behavior. 

## Figures and Tables

**Figure 1 polymers-11-00345-f001:**
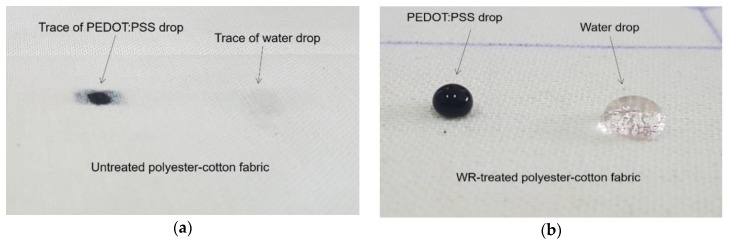
Drops of water and PEDOT:PSS polymer dispersion on (**a**) untreated fabric and (**b**) water repellent (WR)-treated fabric.

**Figure 2 polymers-11-00345-f002:**
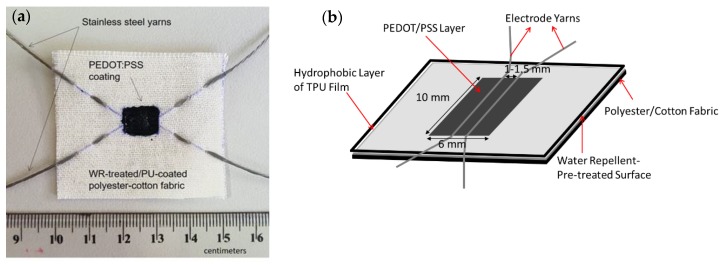
(**a**) Example of the actual textile energy device used in this experiment and (**b**) Schematic design of the textile energy storage device. Note that the electrodes used in the above device were a pair of stainless steel/stainless steel yarns. Other pairs of electrode yarns, such as stainless steel/silver coated polybenzoxazole (Ag-PBO) and Ag-PBO/Ag-PBO, were also used.

**Figure 3 polymers-11-00345-f003:**
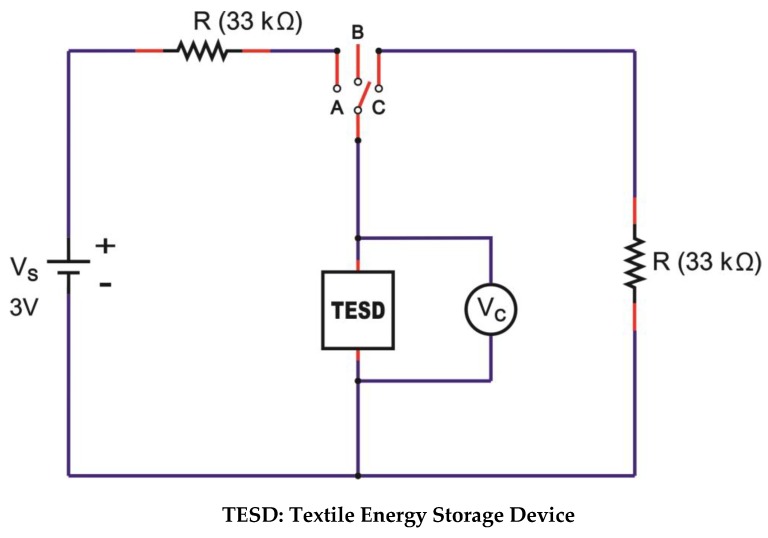
Schematic diagram of charge-discharge measurements set up.

**Figure 4 polymers-11-00345-f004:**
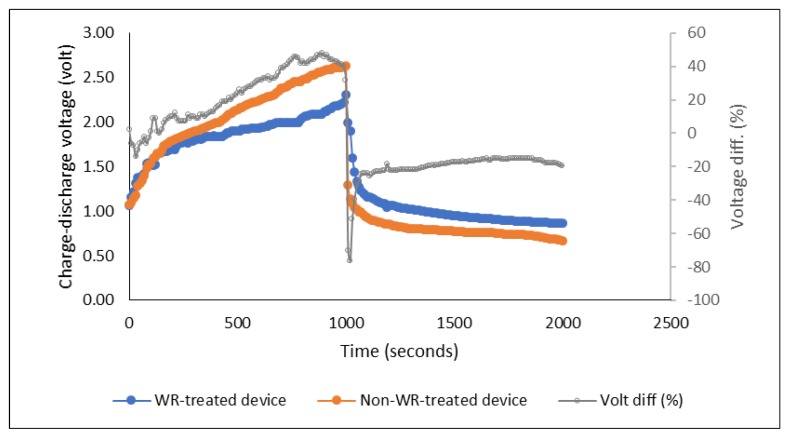
Charge-discharge profile of the water repellent- versus non-water repellent-pre-treated devices (“Volt diff” shows the difference of charge-discharge voltage of the two compared devices recorded along the measured times).

**Figure 5 polymers-11-00345-f005:**
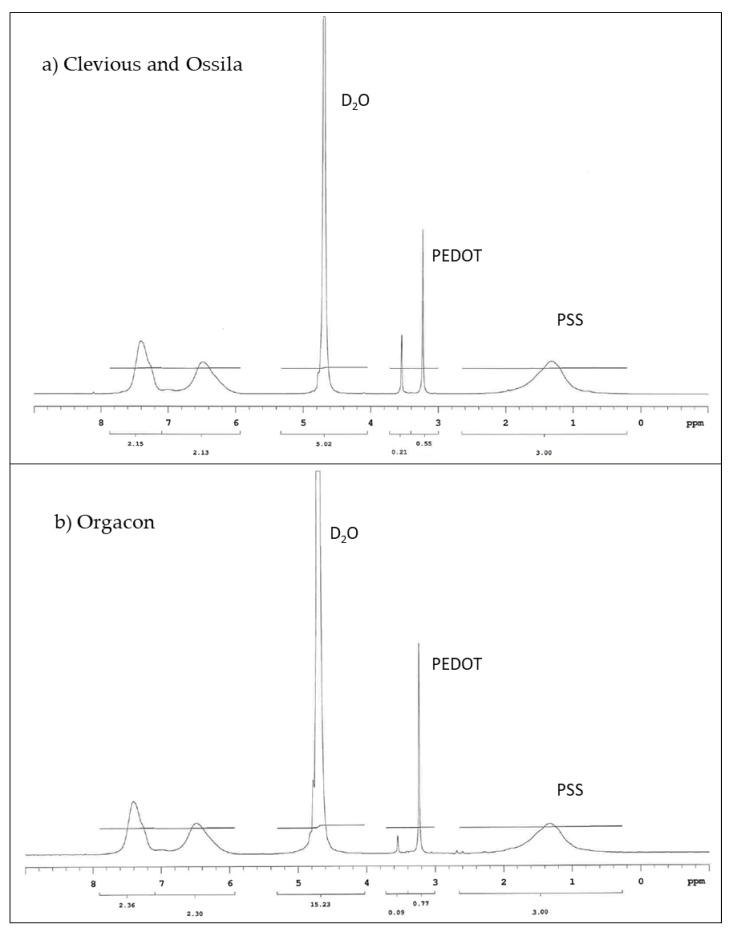
^1^H NMR spectra of PEDOT:PSS (**a**) Clevios and Ossila (**b**) Orgacon.

**Figure 6 polymers-11-00345-f006:**
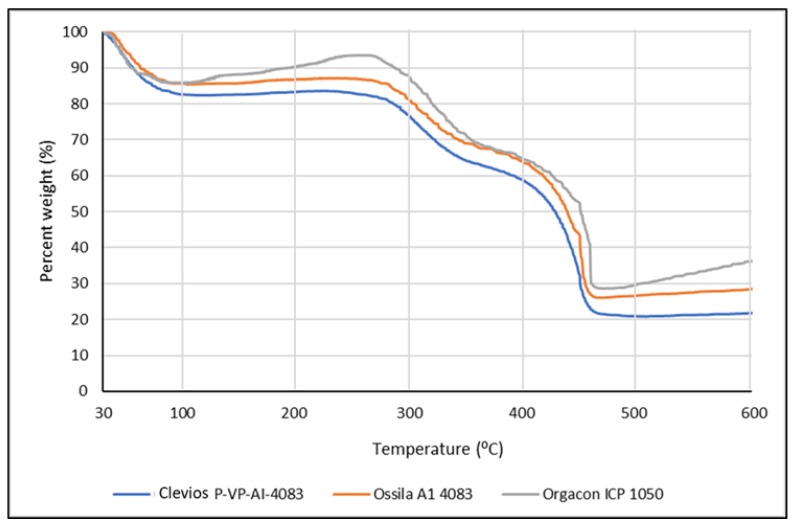
TGA Curves of Clevios, Ossila, and Orgacon PEDOT:PSS.

**Figure 7 polymers-11-00345-f007:**
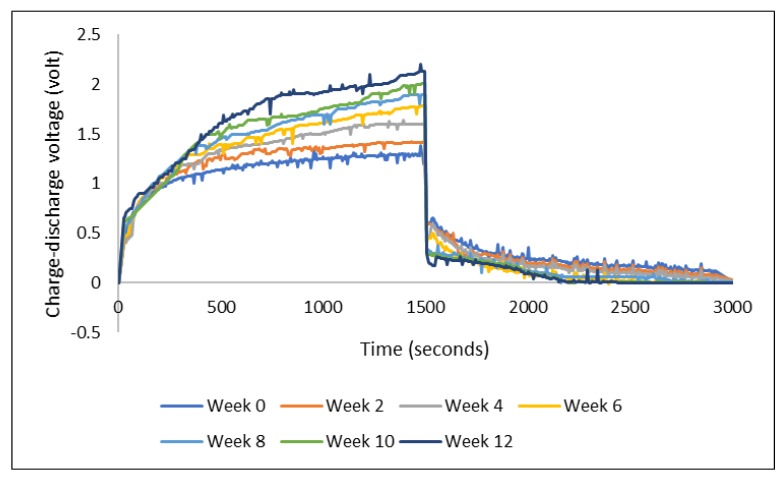
Charge-discharge profiles of a device along the storage in ambient condition.

**Figure 8 polymers-11-00345-f008:**
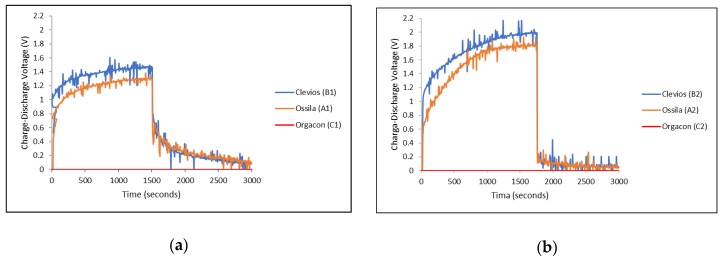
Charge-discharge characteristics of devices of different types of PEDOT:PSS with (**a**) SS/SS electrodes and (**b**) Ag-PBO/Ag-PBO electrodes, each charged at 3 V.

**Figure 9 polymers-11-00345-f009:**
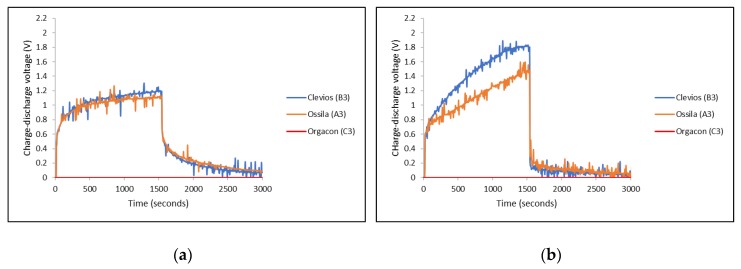
Charge-discharge characteristics of devices of different types of PEDOT:PSS using SS/Ag-PBO electrodes with different polarity of the applied voltage: (**a**) SS(+3 V)/Ag-PBO(0 V) and (**b**) SS(0 V)/Ag-PBO(+3 V).

**Figure 10 polymers-11-00345-f010:**
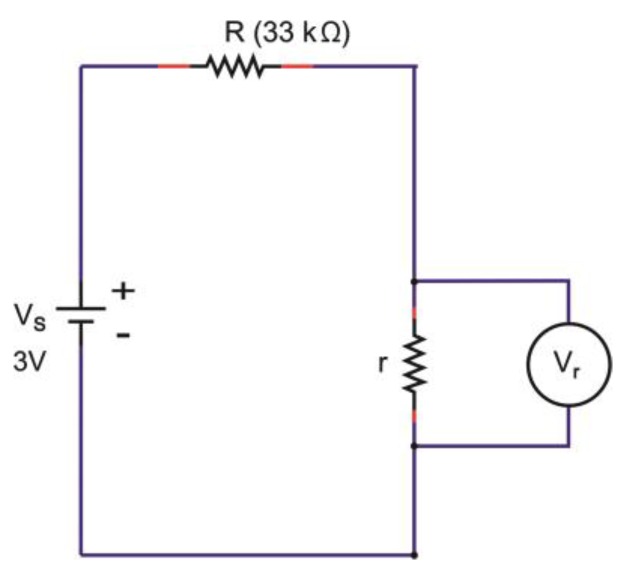
The hypothetical schematic circuit of the Orgacon ICP 1050-containing device with no capacitive behavior.

**Figure 11 polymers-11-00345-f011:**
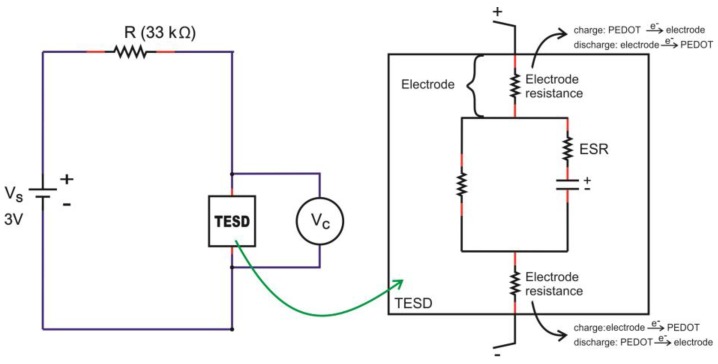
Possible configuration of the electronic system in the TESD cell with capacitive behavior.

**Table 1 polymers-11-00345-t001:** Variation set up of the devices.

Name of Device	Electrodes Pair	Electrolyte
A1	SS/SS	Ossila AI 4083
A2	Ag-PBO/Ag-PBO	Ossila AI 4083
A3	SS/Ag-PBO	Ossila AI 4083
B1	SS/SS	Clevios P-VP-AI-4083
B2	Ag-PBO/Ag-PBO	Clevios P-VP-AI-4083
B3	SS/Ag-PBO	Clevios P-VP-AI-4083
C1	SS/SS	Orgacon ICP 1050
C2	Ag-PBO/Ag-PBO	Orgacon ICP 1050
C3	SS /Ag-PBO	Orgacon ICP 1050
Blank 1	SS/SS	-
Blank 2	Ag-PBO/Ag-PBO	-
Blank 3	SS/Ag-PBO	-

**Table 2 polymers-11-00345-t002:** PEDOT:PSS physical and chemical characteristics based on measurement (NMR and ICP-OES) and technical information.

Characteristics	Clevios P-VP-AI-4083	Ossila A1 4083	Orgacon ICP 1050
Measurement results **			
PEDOT:PSS molar ratio	1.00:5.26	1.00:5.26	1.00:4.65
PEDOT:PSS weight ratio	1.00:6.92	1.00:6.92	1.00:6.11
Concentration of sulfur (mg/L)	2799 ± 10	2845 ± 10	2120 ± 10
Resistivity, ρ (Ω·cm)	1000	1000	0.25–1
Technical information ***			
Solid content (% in water)	1.3–1.7	1.3–1.7	1.1
Resistivity, ρ (Ω·cm)	500–5000	500–5000	N/A
Resistance (Ω/sq) *	N/A	N/A	<100
PEDOT:PSS ratio (by weight)	1:6	1:6	N/A
Work function (eV)	5.2	5.0–5.2	N/A

* The unit was presented differently as Ω/sq (sheet resistance (*R_sq_*)), where ρ = *R_sq_·d* (resistivity equals to sheet resistance times the layer thickness, and when the thickness *d* is 1, the resistivity equals to resistance). The PEDOT:PSS Orgacon ICP 1050 was classified as a highly conductive grade. ** All the data presented in the upper block was taken experimentally from our measurements. *** All the data presented in the lower block was taken from the technical information provided by each company. The information can also be obtained online [37,38,39]. https://www.heraeus.com/media/media/group/doc_group/products_1/conductive_polymers_1/p/CLEVIOS_P_VP_AI_4083.pdf; https://www.ossila.com/products/pedot-pss; https://www.sigmaaldrich.com/catalog/product/aldrich/739332?lang=en&region=ID.

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
