# Peer review of "Charge-Discharge Characteristics of Textile Energy Storage Devices Having Different PEDOT:PSS Ratios and Conductive Yarns Configuration"

_polymers, 2019, doi:10.3390/polym11020345_

Round 1

Reviewer 1 Report

I am generally satisfied with authors reply

I suggest publishing of this manuscript.

Reviewer 2 Report

All open question and modification made for the manuscript are made. paper can be accepted

This manuscript is a resubmission of an earlier submission. The following is a list of the peer review reports and author responses from that submission.

Round 1

Reviewer 1 Report

The manuscript "Significance of PEDOT:PSS Ratio Towards the  Performance of Textile Energy Storage Devices" shows interesting material but there some goals missing and the title is not reflecting with the content of the manuscript. It is for sure interesting using PEDOT:PSS in textile for energy storage but saying so the reader expect certain results like capacitance measurements. the reviewer recomment if keeping the title to include those measurements.The authors showed beneath characterization only charging/discharging experiments very detailed. It would help if talking about energy storage to show the adhesion of such coatings and in case for textiles: Did the authors made delamination experiments after several washing steps of the coated textile? Additional the charging time is chosen to be 1500 s. Why so long and why not different frequencies applied?

The manuscript itself contains too many figures. The reviewer suggest to organise the manuscript better.

As example Figure 1 is well known and can be transferred to supplementary as well Figure 4 doesn,t really give some new information.

Table 2 the technical information shows no references where the parameter obtained.

Figure 6 the NMR spectras are they really new or shown in other work before? The can be moved to supplementary as well

Figure 8, 9 and 10 showing similiar measurements. In general those charge-discharge properties are obtained from current time curves by integration. The Y geaph shows voltage and is a bit confusing. Anyhow those graphs can be shown in 1 or maximal 2 Figures. Additionally why Figure 11 and Figure 13 showing schematic circuits but which information can be drawn from there? Figure 12 can be shown in supplementary. Why have your measurements so much noise?

Some minor points

Page 9, line 282 add reference after charge transport, line 287 add references

Page 10, line 293 and 296 add references

Page 10, line 316 add references

Page 11, line 338 add references

In general using keywords in abstract they need to appear in abstract as well which is not given for conductive polymer and Ag/PBO, please correct that.

Reviewer 2 Report

The authors investigated the electrical conductivity, sulfur content, resistivity, and the characteristics of Energy storage devices of three types of PEDOT: PSS. However, novelty is weak concerning PEDOT: PSS and interpretation are wrong with the following points. Moreover, the principle of Energy storage devices is unclear, and this manuscript should be rejected.

The authors said that the ratio of PEDOT to PSS is essential. Indeed, the electrical conductivity is large for CLEVIOS â„¢ P VP CH 8000 with PEDOT: PSS ratio 1:20 and CLEVIOS â„¢ PH 1000 with 1: 2.5 It is different. However, it is known that electric conductivity is significantly different between CLEVIOS. â„¢ PH 1000 and CLEVIOS. â„¢ P of the same 1.25. Therefore, it is not determined by weight ratio alone.

Although both PEDOT and PSS contain sulfur, it is unknown how to calculate the weight ratio. Also, although the solution concentration has a wide range according to the product information, it is meaningless even if the concentration of sulfur is given while keeping it unclear.

Since Ossila AI 4083 is manufactured by Heraeus and Ossila is an agent, it should be initially the same, and the comparison has no meaning. Similarly, Orgacon ICP 1050 is AGFA manufacturer, and Aldrich is a distributor.

If the film thickness is not accurately known, the resistivity is not, but it is unknown how to calculate the film thickness.